# A chemical signal in human female tears lowers aggression in males

**Shani Agron**[1,2‡]\*, **Claire A. de March**[3,4‡], **Reut Weissgross**[1,2], **Eva Mishor**[1,2],
**Lior Gorodisky**[1,2], **Tali Weiss**[1,2], **Edna Furman-Haran**[1], **Hiroaki Matsunami**[3,4‡],
**Noam Sobel**[1,2‡]\*

1 The Azrieli National Center for Human Brain Imaging and Research, Weizmann Institute of Science, Rehovot, Israel, 2 The Department for Brain Sciences, Weizmann Institute of Science, Rehovot, Israel, 3 Department of Molecular Genetics and Microbiology, Duke University Medical Center, Durham, North Carolina, United States of America, 4 Department of Neurobiology, Duke University Medical Center, Durham, North Carolina, United States of America

‡ SA and CAM share first authorship on this work. HM and NS are joint senior authors on this work.
\* shaniagron@gmail.com (SA); noam.sobel@weizmann.ac.il (NS)

**Data Availability Statement:** All in-vitro/ behavioral/perceptual data generated during this study are included in this published article and its Supplementary materials. All fMRI data are

## Abstract

Rodent tears contain social chemosignals with diverse effects, including blocking male aggression. Human tears also contain a chemosignal that lowers male testosterone, but its behavioral significance was unclear. Because reduced testosterone is associated with reduced aggression, we tested the hypothesis that human tears act like rodent tears to block male aggression. Using a standard behavioral paradigm, we found that sniffing emotional tears with no odor percept reduced human male aggression by 43.7%. To probe the peripheral brain substrates of this effect, we applied tears to 62 human olfactory receptors in vitro. We identified 4 receptors that responded in a dose-dependent manner to this stimulus. Finally, to probe the central brain substrates of this effect, we repeated the experiment concurrent with functional brain imaging. We found that sniffing tears increased functional connectivity between the neural substrates of olfaction and aggression, reducing overall levels of neural activity in the latter. Taken together, our results imply that like in rodents, a human tear–bound chemosignal lowers male aggression, a mechanism that likely relies on the structural and functional overlap in the brain substrates of olfaction and aggression. We suggest that tears are a mammalian-wide mechanism that provides a chemical blanket protecting against aggression.

## Introduction

Mammals use various bodily media to convey social chemical signals. For example, human social chemosignaling research has focused on sweat [1], and rodent research has focused on urine [2]. Social chemosignaling, however, also extends to media such as feces [3], milk [4], and tears [5–12]. Rodent tear signaling has been studied in 2 contexts: reproduction and aggression. In reproduction, a male-specific peptide secreted from the extraorbital lacrimal gland, named exocrine gland-secreting peptide 1 (ESP1), is transduced by female

available for download at OpenNeuro: https://openneuro.org/datasets/ds004274. The custom code used to process the data collected in this study is available at https://gitlab.com/shaniagron/human-emotional-tears-blocks-aggression.

**Funding:** This work was funded by an ISF grant (714103) awarded to NS, and by support to the Sobel lab from the Rob and Cheryl McEwen Fund for Brain Research. Additional support from National Science Foundation grant 1555919 to HM, National Institute of Health grant DC014423 and DC016224 to HM, National Institute of Health grant K99DC018333 to CAdM. The funders had no role in study design, data collection and analysis, decision to publish, or preparation of the manuscript.

**Competing interests:** The authors have declared that no competing interests exist.

**Abbreviations:** AGQ, Aggression Questionnaire; AIC, anterior insula cortex; APR, aggression provocation ratio; AQ, Autism Quotient; BBR, boundary-based registration; ESP1, exocrine gland-secreting peptide 1; FLAME, FMRIB's Local Analysis of Mixed Effects; fMRI, functional magnetic resonance imaging; HRF, hemodynamic response function; MEM, Minimum Essential Media; OR, olfactory receptor; PFC, prefrontal cortex; PPI, psychophysiological interaction; PSAP, point subtraction aggression paradigm; ratCRP1, rat cystatin-related protein 1; ROI, region of interest; STAI, State Anxiety Questionnaire; TP, temporal pole; VAS, visual analog scale.

V2Rp5-expressing vomeronasal sensory neurons [5]. This triggers signals from the accessory olfactory bulb to hypothalamic and amygdaloid nuclei, which enhance female sexual receptive behavior [6]. The tear-bound signal ESP1 is also the primary signal in the Bruce effect, where a pregnant mouse will miscarry upon perceiving an ESP1 signal from a male who did not father the pregnancy [7]. These tear-bound signals function not only within species but also across species: Like ESP1 in mice, rat cystatin-related protein 1 (ratCRP1) is released from male rat tears and alters behavior in female rats. This same rat signal, however, also triggers predator avoidance in mice [9]. Beyond reproductive signaling, a primary domain for rodent tear signaling is aggression. The above noted tear signal ESP1 that promotes sexual behavior in females also increases aggressive behavior in males smelling their own ESP1 secretions [8]. However, most aggression-related tear signaling appears to block rather than promote aggression. This was first identified in blind mole rats, where subordinate males cover themselves in tears to reduce dominant male aggression toward them [10]. Similarly, mice pups emit in their tears exocrine gland-secreting peptide 22, which through a vomeronasal accessory olfactory pathway, reduces male sexual aggression toward them [11]. Finally, female mouse tear liquid contains signals that abolish intermale aggression by modulation of activity in aggression brain networks [12]. In contrast to this extensive body of research into rodent tear chemosignaling, there is only limited evidence for human tear chemosignaling. Human female tears contain a perceptually odorless chemical signal that when sniffed, lowers testosterone in human males [13,14], but the behavioral significance of this effect remains poorly understood. More specifically, one study found that sniffing tears drove a small but significant reduction in ratings of sexual arousal attributed to pictures [13], and the second study observed that despite significantly lowering testosterone, sniffing tears did not affect appetite [14]. Given that reduced male testosterone is associated with reduced male aggression [15], here, we set out to test the hypothesis that like in rodents, human tears contain a chemical signal that blocks aggression. Notably, there are indeed several instances of chemical signals altering hormonal-dependent behavior in humans [16]. Examples include maternal behavior [17,18], ingestive behavior [19,20], social behavior in general [21,22], and sociosexual behavior in particular [23–25]. In other words, that a chemical signal can alter human behavior is not unusual. Moreover, particularly emotional behaviors are a prime candidate for modulation by chemical signals [26], possibly a reflection of their shared neural substrates in the amygdaloid complex [27,28] and an extensive associated brain network spanning ventral temporal cortex, frontal cortex, anterior cingulate cortex, and insula striatum [29]. Given this neural link, and that human aggression can be measured behaviorally using various standardized tasks [30], we set out to measure the aggressive behavioral and brain response following sniffing emotional tears.

## Results

### Sniffing human emotional tears blocks male aggression

In Experiment 1, we asked whether sniffing human emotional perceptually odorless tears reduces aggression in men as it does in male rodents. First, we harvested emotional tears from human female donors (6 regular donor women, age range 22 to 25 years) using methods previously described [13] (see Methods). Because tears that trickled down the cheek and into the collection device may have collected skin-bound signaling molecules not originating from tear fluid, as a control substance, we trickled saline down the cheeks of the very same donors and collected it in a similar manner. Next, we used the point subtraction aggression paradigm (PSAP), a validated measure of aggression in response to provocation [31,32]. In brief, in the PSAP, participants play a monetary game with an opponent they are told is human but is, in

fact, a computer algorithm. The game contains provocation events where money is "unfairly" taken from the participant, and revenge events, where the participant can deduct money from his opponent at no personal gain. Aggression is estimated by the aggression provocation ratio (APR), namely, the ratio between the number of revenge responses to the number of provocations the participant experienced. A higher APR reflects higher aggression. Before the PSAP, each participant went through a stimulus exposure procedure. Participants were told they are sniffing subthreshold concentrations of odors, but it was not stated at this stage what they were (they provided advanced consent for "assorted odors, including body odors"). A sniff jar containing 1 ml of stimulus was presented before the participant's nose 13 times, with an approximately 35-second intersniff interval (S1A Fig). The first 3 sniffs were of saline solution (blank), and the following 10 sniffs were of the stimulus (either emotional tears or trickled saline). After each sniff, the participants used a visual analog scale (VAS) to rate the pleasantness, intensity, and familiarity of the stimulus. After this, a pad containing 100 μl of the stimulus (tears/trickled saline) was secured to the participant's upper lip facing out, keeping the participant continuously exposed to the stimulus for the duration of the experiment. Participants (we recruited 31 but retained 25 men for analysis, age = 25.84 ± 3.46; see Methods for exclusion criteria) came to the lab on 2 consecutive days, at the same time of day, and engaged in a PSAP game, once after sniffing tears and once after sniffing trickled saline, counterbalanced for order, double-blind for condition. Consistent with previous results, we observed no perceptual differences between tears and trickled saline, which did not significantly differ in perceived intensity, pleasantness, and familiarity (Stimuli: $F_{(1,20)} = 2.53$, $p = 0.127$; Descriptor: $F_{(1,40)} = 1.183$, $p = 0.317$; Order: $F_{(1,20)} = 1.665$, $p = 0.212$, with no interactions) (Figs 1A–1C and S2A–S2C and S1 Data). Moreover, both stimuli did not perceptually differ from saline solution (blank), emphasizing the perceptually odorless nature of the stimulus (Blank versus Trickled Saline: $F_{(1,21)} = 1.815$, $p = 0.192$, Descriptor: $F_{(2,42)} = 1.735$, $p = 0.189$; Blank versus Tears: $F_{(1,21)} = 0.0004$, $p = 0.984$, Descriptor: $F_{(2,42)} = 1.476$, $p = 0.24$, without order effect or interactions for both stimuli) (S2D–S2F Fig).

In turn, we observed a remarkable reduction in aggression following exposure to tears. Whereas mean APR ± SD following trickled saline was 1.67±1.7, APR following tears was 0.94±0.92, or in other words, tears drove a 43.7% reduction in aggression (Shapiro–Wilk, W = 0.827, $p < 0.001$, implying a nonnormal distribution dictating a nonparametric test: Wilcoxon signed rank Z = 53, $p = 0.031$, effect size ($r_{rb}$) = 0.541, with no effect of order: Wilcoxon signed rank Z = 98, $p = 0.555$, effect size ($r_{rb}$) = 0.152. If we nevertheless use a parametric approach, the effect remains the same: $t(24) = −2.68$, $p = 0.013$, Cohen's d = 0.527, with no effect of order: $t(24) = −0.548$, $p = 0.59$ (the effect remains even if we include the outlier in the analysis: Wilcoxon signed rank Z = 66, $p = 0.05$, effect size ($r_{rb}$) = 0.48) (Fig 2A and 2B, S1 Data). Finally, to further evaluate the robustness of this effect, we ran a bootstrap analysis. We randomly reassigned paired outcomes 10,000 times in order to generate a random distribution of results and then compared the actual result we obtained to this distribution. We observe that the chance probability to obtain this outcome is 2.9% (Fig 2C). These results suggest that, like in rodents, a primary chemosignaling function of human emotional tears may be a "stop aggression" signal. We next set out to ask whether the main olfactory system can respond to this perceptually odorless message.

## Emotional tears activate specific human olfactory receptors in vitro

To ask if the human olfactory system can process signals from tears, in Experiment 2, we expressed 62 human olfactory receptors (ORs) (S1 Table) in Hana3A cells and monitored their real-time activation by tears or saline using a luciferase-based assay as previously described

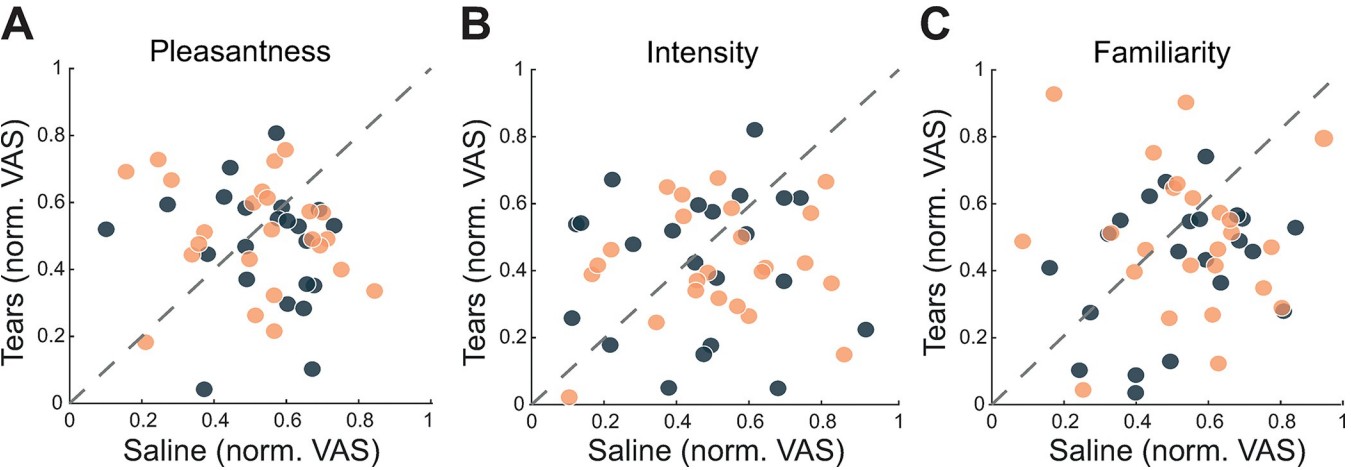

**Fig 1. Tears did not perceptually differ from saline.** Scatter plots of the normalized VAS ratings of tears and trickled saline for (**A**) pleasantness, (**B**) intensity, and (**C**) familiarity. Each dot is the average of 10 sniffs by a given participant; light-colored dots are from Experiment 1 (*n* = 22), and dark dots are from Experiment 3 (*n* = 24). The data in (**A**-**C**) are presented along a unit slope line (X = Y), such that if points accumulate above the line, this implies higher values after tears; if points accumulate below the line, this implies higher values after saline; and if points are distributed around the line, this implies no difference. Data used to generate graphs can be found in S1 Data.

[33,34]. In this initial screening, we observed 21 ORs activated by tears and not by trickled saline (quadruplicates for each receptor type. all T > 2.24, all uncorrected $p <= 0.05$) (S3 Fig and S1 Table and S2 Data). To further probe for a typical sensory response profile in these 21 candidate receptors, we repeated the experiment with 6 serial dilutions of emotional tears (between 1% and 3.16% v/v). This confirmed the OR response in 4 of these 21 ORs: OR2J2, OR11H6, OR5A1, and OR2AG2 (all done in triplicates or quadruplicates, (F(1,28) > 5.827, $p < 0.023$) (Fig 3 and S3 Table and S3 Data). In other words, human emotional tears, although perceptually odorless, activate specific human ORs in vitro, and this may provide the molecular basis for human social chemosignaling through tears. Having verified that this stimulus has pronounced impact on behavior and the potential for generating a response through the main human olfactory system, we next set out to ask how this is reflected in the brain.

## Sniffing tears coordinates the brain response in reactive aggression

To gauge the brain response to sniffing tears in the context of aggression, in Experiment 3, we performed functional magnetic resonance imaging (fMRI) in participants playing the PSAP in the MRI scanner (we recruited 33 but retained 26 men for analysis, age = 27 ± 3.2; see Methods for exclusions), day after day, once exposed to tears and once to trickled saline, double-blind for condition (i.e., 52 scans in total). Again, we observed no perceptual difference between trickled saline and tears (Stimuli: F(1,22) = 1.4, $p = 0.25$; Descriptor: Mauchly's sphericity test $p < 0.5$, Huynh–Feldt correction F(1,44) = 2.283, $p = 0.124$; Order: F(1,22) = 1.117, $p = 0.3$, with no interactions) (Figs 1A–1C and S2A–S2C). In turn, in this challenging experiment, the behavioral effect of tears on aggressive behavior was only subtle. The mean APR ± SD following trickled saline was 1.306 ± 1.6, and APR following tears was 0.967 ± 1.357 (Shapiro–Wilk, W = 0.857, $p < 0.002$, implying a nonnormal distribution dictating a nonparametric test). Given the results of Experiment 1 where tears significantly reduced aggression, we apply a one-tailed hypothesis: Wilcoxon signed rank Z = 67, effect size ($r_{rb}$) = 0.42, $p = 0.048$, one-tailed (Fig 2D and 2E and S1 Data). We think that this weaker (effect size of 0.541 in Experiment 1 versus effect size of 0.42 in Experiment 3) result reflects the psychological dynamics of

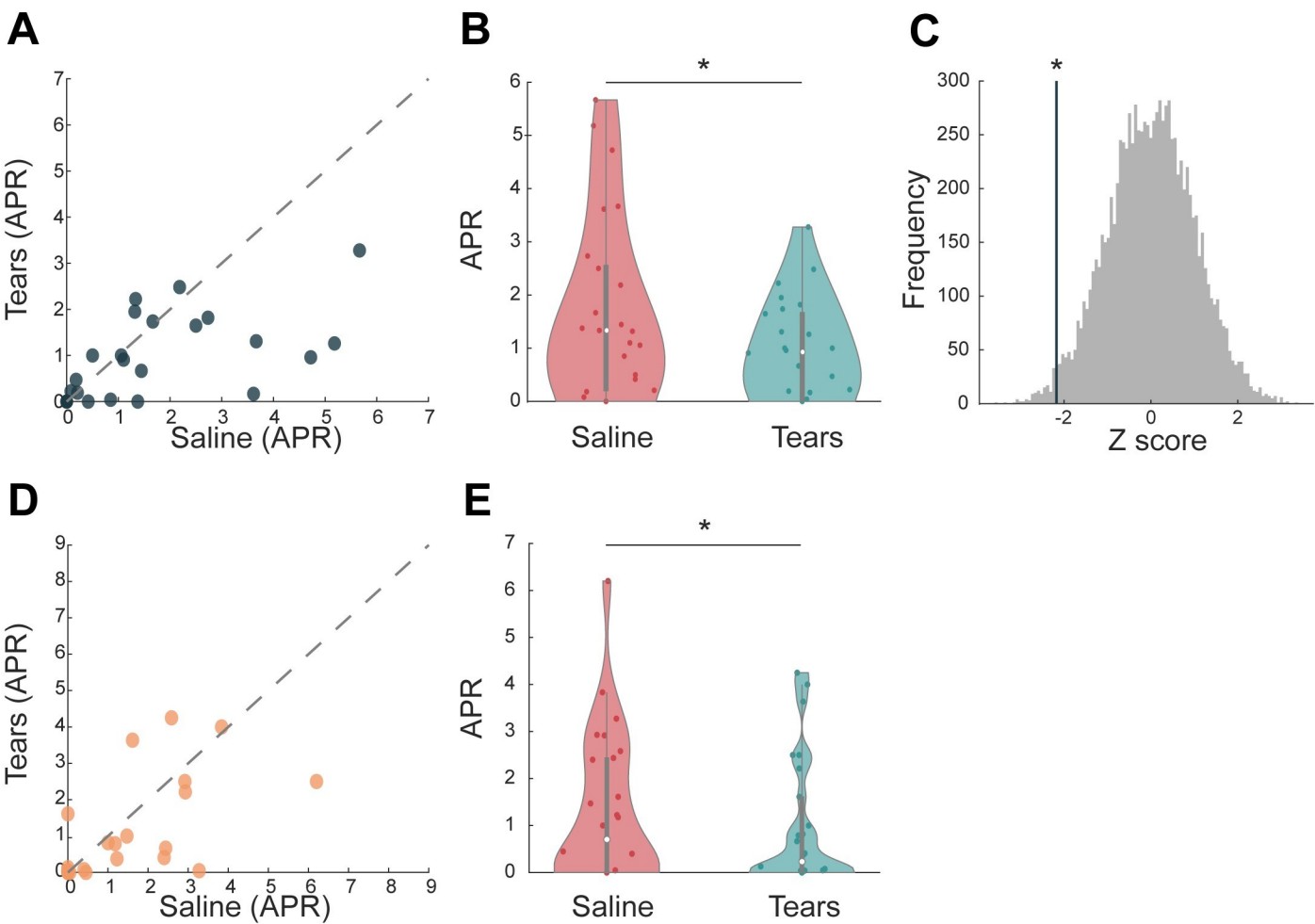

**Fig 2. Sniffing emotional tears blocks male aggression.** (**A**) Aggression ratings (APR) in Experiment 1, obtained after exposure to tears or saline. Each dot is a participant, *n* = 25. (**B**) The same data as in (**A**), presented in violin-plot. Each dot is a participant. The white dot represents the median, and the gray bar represents the quartiles. Saline in red and tears in blue. (**C**) Bootstrap analysis. Gray lines represent the 10,000 repetitions; the blue line represents the actual APR difference between saline and tears. (**D**) Scatter plots of the aggression ratings (APR) obtained in the MRI (Experiment 3) after exposure to tears or saline. Each dot is a participant, *n* = 26. (**E**) The same data as in (**A**) presented in violin-plot. Saline in red and tears in blue. The data in (**A**) and (**D**) are presented along a unit slope line (X = Y), such that if points accumulate above the line, this implies higher values after tears; if points accumulate below the line, this implies higher values after saline; and if points are distributed around the line, this implies no difference. Data used to generate graphs can be found in S1 Data.

the day-after-day MRI experiment, as participants were more aggressive on the second day regardless of condition. We detail this in S4 Fig.

We next explored the brain response to provocation under tears versus saline. We generated a whole-brain voxel-wise statistical parametric map ($p < 0.005$, cluster-corrected for multiple comparisons). Provocation versus inactive time regardless of condition revealed a typical salience network activation, which included typical provocation-associated regions [35] such as the right inferior and middle frontal gyri (S5 Fig) (see S3 Table for full list of regions). This pattern suggests that we effectively recruited the neural substrates of aggression typically activated in this task. In turn, the ANOVA contrast of provocation with the added interaction level of saline versus tears ($p < 0.005$, cluster-corrected for multiple comparisons) revealed no region where provocation under tears was associated with a significant increase in activity, but several brain structures where provocation under tears was associated with a significant reduction in activity (S4 Table for full list of regions). Notably, dampening rather than increasing

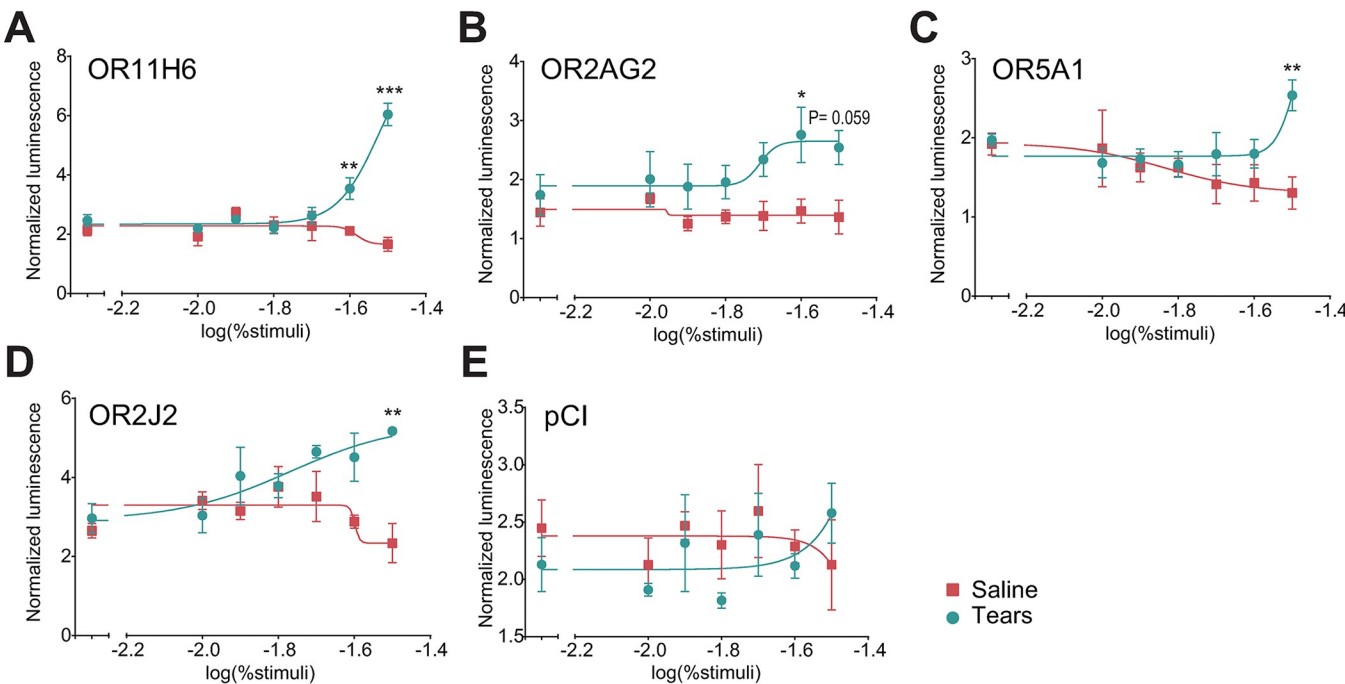

**Fig 3. Perceptually odorless emotional tears activated human olfactory receptors in vitro.** The normalized luminescence from the OR response to tears or trickled saline, ranging in concentration from 1% to 3.16% (in CD293 simulation medium). A dose response to tears but not trickled saline was evident in receptors (**A**) OR11H6, (**B**) OR2AG2, (**C**) OR5A1, and (**D**) OR2J2. (**E**) No dose response was seen in the control empty vector—pCI. Each dot is the mean of 3 repetitions for either tears (blue) or saline (red), and the error bar is the standard error across 3 replications. A two-way ANOVA followed by a Sidàk's multiple comparison test was performed at each concentration between the OR response to tears and saline (*** = $p < 0.0001$, ** = $p < 0.001$, * = $p < 0.05$, no symbol = not significantly different). Data used to generate graphs can be found in S3 Data.

activity by tears is consistent with the one previous functional neuroimaging study we conducted with tears [13], and the extent of this effect here is convincing considering the very strict criteria applied, namely, a significant interaction after correcting for multiple voxel-wise comparisons. Out of these regions where tears had this dampening impact, 2 regions have been repeatedly implicated in aggression [35]: the left anterior insula cortex (left AIC) [36] and bilateral prefrontal cortex (PFC) [37] (Fig 4A). Tellingly, we observe that the difference in the beta values between conditions (tears and saline) in these regions was significantly correlated with the difference in the level of aggression expressed in the scanner as measured by APR (left AIC Spearman rank correlation: r = 0.54, $p$ = 0.006, PFC Spearman rank correlation: r = 0.41, $p$ = 0.046) (Figs 4B, 4C, S6D, and S6E) (S1 Data). These correlations suggest that we captured a parametric link between brain and behavior, whereby tears are associated with dampening provocation-induced activity in the brain aggression network. We observe that although we balanced our study for order, the later subject exclusions violated this balance. To assure that these brain activity patterns were not a result of this imbalance, we conducted 2 analyses. We have 9 fMRI participants who had saline-first. Thus, a balanced group from this perspective is reduced to 18 participants, which is borderline in power. To overcome this, using a bootstrap approach, we randomly selected balanced groups of 18 participants 10,000 times, and each time conducted the analysis to create a distribution of beta values in the left AIC and PFC. The reduced activation in these regions remained significant (left AIC: mean t (17) = 2.417, mean $p$-value = 0.036, and for the PFC: mean t (17) = 3.814, mean $p$-value = 0.0015 (S6A and S6B Fig). Second, we conducted a whole brain analysis on a group of 18 participants balanced for order. We removed 6 tears-first participants by removing those with a lower Aggression

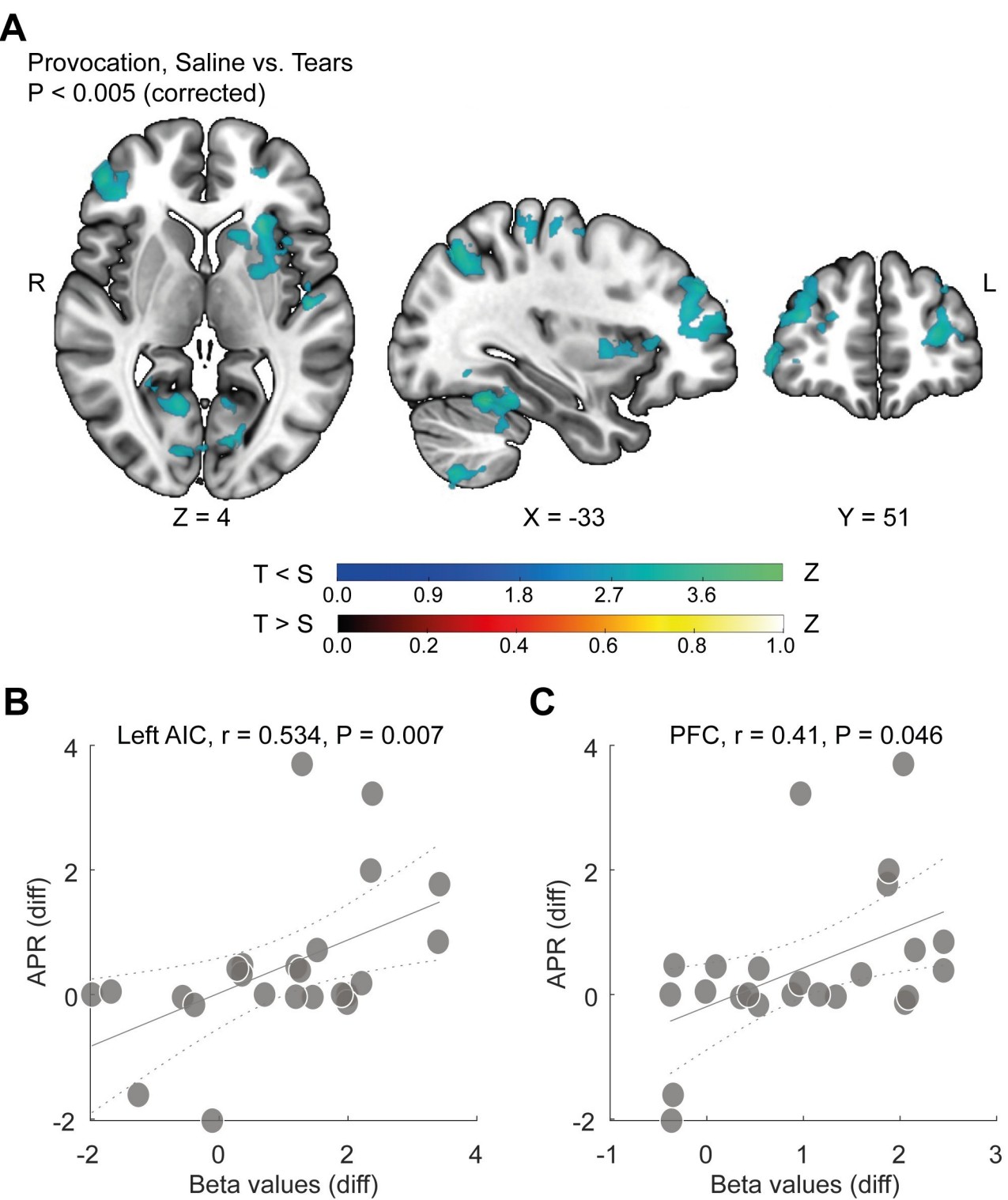

**Fig 4. Tears reduced activation in the brain substrates of reactive aggression.** (**A**) Statistical map of the GLM ANOVA Provocation > inactive time contrast with an added level of saline vs. tears (tears < saline in blue; tears > saline in red), *n* = 24. GLM z threshold > 2.58, cluster corrected to *p* = 0.05. Color bars represent z-values. (**B**, **C**) Correlation between differences in behavioral APR scores (saline -tears) and differences in beta values (saline- tears) of (**B**) left AIC and (**C**) PFC. Each dot represents a participant, *n* = 24. The continuous line represents the fit. The dashed line marks the confidence bounds. Spearman rank correlation coefficient and *p*-values are depicted. Data used to generate graphs can be found in S1 Data; fMRI data are available at https://openneuro.org/datasets/ds004274.

Questionnaire (AGQ) score, to create 2 groups balanced in this respect (mean AGQ score for: saline first group = 35.89 ± 11.73, tears first group = 46.67 ± 14.4, $p$ = 0.1). Once again, we observed the same pattern of reduced brain response as in the larger group (z threshold > 2.31, refracting a $p < 0.01$ corrected for multiple comparisons), and the correlation between behavior and brain response in the left AIC and PFC was also maintained (S6C–S6E Fig).

Next, to investigate how these regions may be modulating aggression under tears, we probed their functional connectivity with the entire brain under tears versus saline. We applied whole-brain psychophysiological interaction (PPI) analysis [38] using the left AIC and PFC functional regions of interest (ROIs) as seeds ($p < 0.005$, cluster-corrected for multiple comparisons). We observed that tears significantly impacted functional connectivity only for the left AIC, which under tears significantly increased connectivity specifically with the right temporal pole (right TP) extending into the right amygdala and piriform cortex (Fig 5). These brain regions share structural connectivity and constitute a functional brain network repeatedly implicated in olfaction [39] and aggression [40]. We further observe that the greater the difference in aggression between tears and saline, the greater the increase in connectivity associated with tears between the left AIC and right amygdala (Spearman rank correlation: r = 0.407, $p$ = 0.048) (Fig 5D) (S1 Data). We did not, however, observe such a link with the right TP (Spearman rank correlation: r = 0.26, $p$ = 0.217). In conclusion, tears significantly increased functional connectivity within a network of brain structures associated with aggression and olfaction, and this increase was correlated with the individual behavioral impact of sniffing tears. Combining the 2 imaging results, (1) that tears reduce provocation-related activity in the neural substrates of reactive aggression and (2) that tears increase functional connectivity between the neural substrates of reactive aggression and the neural substrates of olfaction, we conclude that tears coordinate the brain aggression response.

## Discussion

Charles Darwin was particularly puzzled by the behavior of human emotional tearing, and for lack of apparent function beyond ocular maintenance, he concluded that weeping is "an incidental result" [41]. However, a large body of data has since convincingly demonstrated that tears do have a role beyond ocular maintenance in that they serve mammals as a social chemosignaling media that can be emitted on demand. As detailed throughout this manuscript, this has been best documented in rodents [5–12]. Moreover, a recent study found that dogs also shed emotional tears and that these are visually perceived by humans [42]. That study did not ask whether humans also chemically perceive dog tears, but humans clearly chemically perceive the tears of other humans. In rodents, the chemosensing of tears involves the accessory olfactory system [5–12]. Humans, however, don't have an accessory olfactory system [43,44]. Instead, here, we find that perceptually odorless emotional tears activate 4 specific human main ORs in a dose-dependent manner. We also find that this signal reduces overall levels of activity in the aggression brain network and increases functional connectivity between the brain substrates of olfaction and aggression. Finally, we find that sniffing this signal is associated with a remarkable 43.7% reduction in aggressive behavior. This study did not examine the ecological aspects of this effect, so we can only speculate as to its role in human behavior. We note that crying often occurs in very close-range interactions, to the extent that "kissing teary cheeks" is a recurring theme across cultures. Thus, chemosensing of tears is a viable possibility in human behavior. Moreover, although we tested tears from women donors, we speculate that all tears would have a similar effect. This becomes particularly ecologically relevant with infant tears, as infants lack verbal tools to curb aggression against them and are therefore more likely to rely on chemosignals.

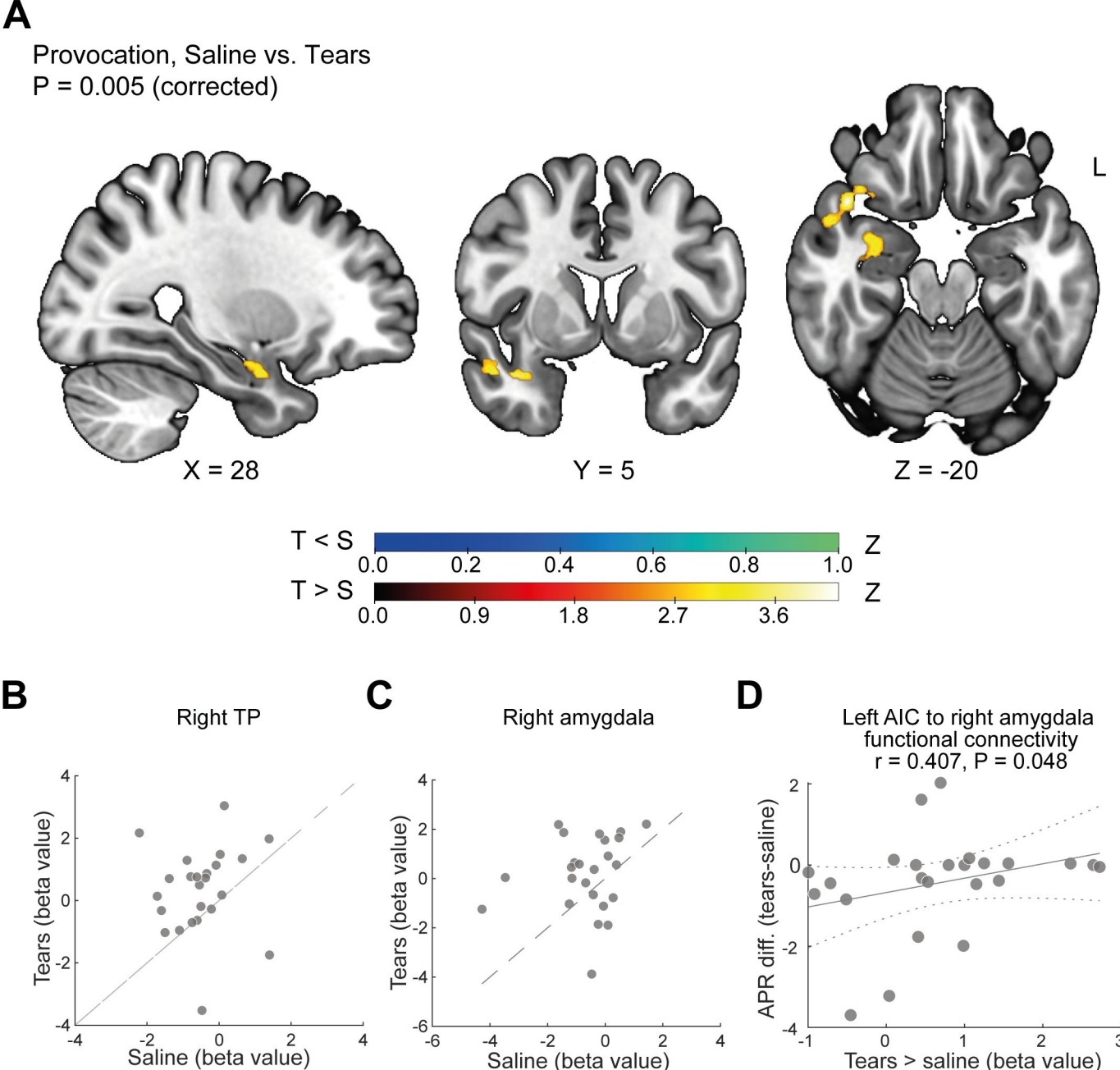

**Fig 5. Tears coordinate the brain response in aggression.** (**A**) Functional connectivity statistical parametric map during provocation > inactive time with an added level of saline vs. tears. Tears < saline in blue. Tears > saline in hot colors. Color bars represent z-values, *n* = 24. (**B**, **C**) Scatter plots of tears vs. saline present the beta values of functional connectivity between left AIC and (**B**) right TP and (**C**) right amygdala. Each dot represents a subject, *n* = 24. The data are presented along a unit slope line (X = Y), such that if points accumulate above the line, this implies higher beta values for tears; if points accumulate below the line, this implies higher beta values for saline; and if points are distributed around the line, this implies no difference. (**D**) Spearman rank correlation between the difference in APR scores in tears vs. saline and increase in connectivity between the left AIC and right amygdala under tears. Each dot represents a participant, *n* = 24. The continuous line represents the fit. The dashed line marks the confidence bounds. Spearman rank correlation coefficient and *p*-value are depicted. Data used to generate graphs can be found in S1 Data; fMRI data are available at https://openneuro.org/datasets/ds004274.

The brain mechanisms of olfaction and the brain mechanisms of aggression are highly overlapping [45]. Indeed, the increased connectivity in aggression under tears generated an image (Fig 4A) that resembles typical olfactory activations [39], with increases in the TP and

amygdala–piriform. In other words, the neuroanatomical overlap between olfaction and aggression places olfactory stimuli in a privileged brain setting for modulation of aggression. This is well known in rodents [45] and even insects [46], and our results imply the same in humans. Our results, combined with previous results on a separate body odor that reduces aggression [47], and a brain response to the body odor of aggression [48,49], together implicate social chemosignaling as a meaningful factor in human aggressive behavior. Moreover, given the role of aggression in social interaction, this olfaction–aggression link may help explain the impaired sociality sometimes evident in human anosmia [50,51]. Finally, the olfaction–aggression link also highlights what we view as a particular strength of the current imaging results, and that is the correlations that emerged between MRI and behavior. This relation was evident in the deactivations associated with tears in the left AIC and PFC, and in the connectivity of the left AIC with the right amygdala. These correlations imply that we captured a parametric brain–behavior interaction.

Whereas the above is a particular strength, this study also had several weaknesses we would like to acknowledge. First, we identify 4 of 62 human OR subtypes that respond to tears. We tested 62 because that is the number we had validated, but humans have approximately 350 receptor types [52], so the actual number is likely greater than 4 and remains unknown. Moreover, that these receptors respond does not prove that these receptors are responsible for the effects we observed, a limitation difficult to overcome in a model where we cannot generate a knockout. A second limitation we would like to acknowledge is that we didn't test for the effect of sniffing tears in women. Not doing this here stems from the overwhelming complication and cost (in time) of collecting the stimulus. We used nearly 1.5 ml of tears per participant per experiment. In other words, this study entailed at least 125 donation sessions where we collected over 160 ml of emotional tears. We hypothesized that male participants were a good place to start because the link between testosterone and aggression is clearer in men than in women [15]. However, given the sexual dimorphism in brain processing of social chemosignals [8,47,53,54], we need to conduct a similar study in women in order to obtain a full picture on the role of this chemosignal in human behavior. A third limitation we would like to acknowledge is that the behavioral effect of tears was reduced in the scanner setting. This is perhaps unsurprising given the discomfort of our participants on scanner day 2, but this remains a limitation. Finally, we note that given the likely hormonally mediated response we are measuring, we expected to observe tear-related alterations in activation or connectivity in the hypothalamus, yet these did not emerge, and we consider this a concern.

Despite the above limitations, we nevertheless reach at several strong conclusions in this study; primarily that a perceptually odorless social signal activates primary ORs, altering activity and connectivity in an olfaction–aggression brain network, all in association with significant shifts in aggressive behavior. This depicts tears as a chemical blanket protecting against aggression, a mechanism common to rodents and humans.

## Methods

### Tear collection for all experiments

We conducted a long-term screen for women who can cry with ease, ultimately identifying 6 regular donor women (age range 22 to 25 years) who participated after providing written informed consent to procedures approved by the Weizmann Institute IRB (Protocol #1597–1). Participation was recurring, 1 donation per day, with a total average of 15.5 ± 9.7 donation-days per donor, and an average of 1.6 ± 0.2 ml per donation. Donors completed general detailed demographic questionnaires upon enlistment and then specific questionnaires with each donation, on questions such as menstrual phase, dietary consumption around the day of

donation, and the nature of the emotions during donation. All donors were under combined hormonal contraceptives to eliminate the possible effects of ovulation on body odor. Donors were instructed to remove any cosmetics on the evening before donation day and to avoid cosmetics until after donation. To obtain tears, the donor women watched sad film clips in isolation and used a mirror to place a vial and capture the tears trickling down their cheeks. A typical donation used in this study contained approximately 1.6 ml of tears. Before tear collection, saline was trickled down the cheek of the donor woman and collected in the same way as tears (trickled saline). Both tears and saline were flash-frozen in liquid nitrogen after collection and kept at −80˚C until use. Upon use, tears were thawed at room temperature. In Experiments 2 and 3, the thawed stimulus was incubated at 30˚C for approximately 10 minutes to achieve temperature equilibrium and a headspace in the sniff jar.

## Experiment 1

**Participants.** To estimate the needed sample size, we relied on previous work measuring chemosignal effects on aggressive behavior [47] that included 25 men. Therefore, 31 healthy men (mean age = 26.2 ± 4.1) with no chronic use of medication were recruited for the experiment. All participants provided written informed consent to procedures approved by the Weizmann Institute IRB committee (Protocol #1872–1) and completed a demographics questionnaire.

**Procedures.** The experiment was conducted in a stainless steel–coated odor-nonadherent room built specifically for human olfaction experiments. Participants came to lab on 2 consecutive days, at the same time of day. Each participant was tested once with tears and once with trickled saline, counterbalanced for order across participants, and double-blind to compound identity. Coauthor RW was solely responsible for double-blinding. She arbitrarily marked the stimuli as "A" and "B" and provided these arbitrarily marked stimuli to lead author SA for continued experimentation. Labels were unblinded only after analysis. Consistent with previous recommendations on consistent experimenter-to-participant gender in human chemosignaling studies [55], all experimenters were women. First, participants were exposed to the stimuli in 13 timed sniffs from a glass jar (27 × 57 mm) containing 1 ml of stimulus, with an intersniff interval of approximately 35 seconds. The first 3 sniffs were of saline solution (blank), and the following 10 sniffs were of experimental compound (either tears or trickled saline). After each sniff, participants rated odor intensity, pleasantness, and familiarity on a visual analog scale (VAS). Next, a pad containing 100 μl of impregnated stimulus was pasted under the participant's nostrils, facing out, for continuous stimulus exposure. Immediately afterward, participants engaged in the point subtraction aggression paradigm (PSAP). The PSAP is an online computer game that participants play against a fictitious opponent who they are led to believe is a real person [31,32]. Participants are told that the goal of the game is to earn as much money as possible, and they actually receive this money at the end of the game. Before the game, participants were told that a random assignment will put one player in the position of the ability to "steal" money from the other player, yet the other player will be only in the position of the ability to deduct money from the other player, but at no personal gain. A fictitious lottery always placed the participant in the latter position. In our version of the PSAP, participants are equipped with 2 squeeze balls, one in each hand. Simultaneous press of both balls for a duration of 5 seconds earns the participant 1 NIS (1 NIS = approximately $0.33). The game is parsed into "events," and each monetary event lasts 10 seconds, enabling the participant to gain 2 NIS. Occasionally, the participant will notice that his acquired sum of money is suddenly reduced. He is led to understand that this is because his opponent took money from him. These are provocation events. The participant has 2 alternatives in response: one is to

disregard this and continue as usual, and the other is to deduce 3 NIS from the opponent, but at no personal gain. Such deductions are actuated by pressing 1 squeeze ball rather than 2. This is considered a revenge event and is consistent with the definition of aggression [56], whereby aggression is any act intended to harm another individual who would rather avoid such treatment. The harm need not be physical (e.g., can be financial) but must lead to some aversive consequence for the recipient. In the PSAP, aggression is estimated by the aggression provocation ratio (APR), namely, the ratio between the number of revenge responses to the number of provocations the participant experienced. A higher APR reflects higher aggression. Participants donated saliva before the exposure to the stimulus, and after the PSAP game, however, due to the COVID-19 outbreak, saliva samples were not analyzed. All Experiment 1 procedures lasted between 90 and 120 minutes per participant per day.

**Questionnaires.** At the end of each day, the participants were asked to fill out a questionnaire to rate their desire to meet their opponent (S7 Fig). They were also given the opportunity to share their thoughts on the experiment and their opponent using their own words. This was done so that we could identify and exclude any participants who understood that they were not playing against a real person. At the conclusion of the experiment, the participants completed the State Anxiety Questionnaire (STAI) [57] and the Buss and Perry Aggression Questionnaire (AGQ) [58]. (S8 Fig). Because social chemosignaling may be altered in autism, we asked participants to complete the Autism Quotient (AQ) questionnaire [59] to monitor for autistic traits.

**Statistical analysis.** All data were analyzed using MATLAB R2019b (MathWorks) and JASP (version 0.15). Odor ratings were normalized to min-max values for each participant based on both stimuli ratings. Each analysis began with an estimation of data distribution using the Shapiro–Wilk normality test. Normally distributed data were analyzed using ANOVA followed by planned $t$ tests (two-tailed for nondirectional hypotheses, and one-tailed for directional hypotheses). When data did not distribute normally (Shapiro–Wilk $p < 0.05$), we used nonparametric analysis methods such as Wilcoxon signed rank test (for paired data), premutation test (to test for order effects), or Wilcoxon sum rank test (for independent samples). In all cases, we then also report a parametric analysis for reference.

**Exclusions.** A total of 6 participants were excluded: 5 participants due to technical faults of the acquisition system, particularly the squeeze balls that failed for a window of time. One participant was defined as an outlier by the interquartile range of APR scores (< Q1-1.5 or > Q3+1.5).

## Experiment 2

**Olfactory receptor activity assay.** Hana3A cells were cultured in Minimum Essential Media (MEM) supplemented with 10% fetal bovine serum, with penicillin–streptomycin and amphotericin B at 37˚C, saturating humidity, and 5% $CO_2$. The Dual-Glo luciferase assay (Promega) was used to determine OR activation by monitoring the activity of Firefly and Renilla luciferase in Hana3A cells, as previously described [60]. Briefly, firefly luciferase, driven by a cAMP response element promoter (CRE-Luc, Stratagene), was used to determine OR activation levels, and the constitutively produced Renilla luciferase (SV40-RL) was used to normalize the luciferase activity in each well. For each well of a 96-well plate, 5 ng of SV40-RL, 10 ng of CRE-Luc, 5 ng of human RTP1S [33], 2.5 ng of M3 muscarinic receptor [46], and 5 ng of Rho-tagged OR plasmid DNA (or empty vector pCI) were transfected 18 to 24 hours before tear or saline stimulations. The stimuli (tears and trickled saline) were diluted in CD293 media supplemented with copper and glutamine (CD293 stimulation medium) to the desired final concentration (% in v/v), and 25 µl of the stimulation solution was injected into each well and

incubated at 37°C, 5% $CO_2$ for 3½ hours. Firefly and Renilla luciferase luminescence was then recorded following the manufacturer's protocol on a POLARstar OPTIMA plate reader (BMG Labtech). Data were analyzed using Microsoft Excel and GraphPad Prism. Normalized activity for each well was further calculated as (Luc-400)/(Rluc-400) where Luc = luminescence of firefly luciferase, Rluc = Renilla luminescence, and 400 corresponds to the luminescence of an empty well. In the OR activation screening, the stimuli were diluted in CD293 medium to reach a concentration of 3.16% (v/v). This concentration does not lead to a non-OR-specific effect on cells (S9 Fig). The normalized luminescence induced by the stimulus was compared to that of CD293 medium, $n = 4$ for each receptor, for the screening. Activation was determined by a one-tailed paired $t$ test with a $p$-value $< = 0.05$. (see the full list of OR and obtained $p$-values in S1 Table). Activation was next verified by a dose–response assay of 6 serial dilutions ranging from 1% to 3.16% (v/v) and CD293 medium (0% stimulus) for baseline. OR responses to tears and saline were analyzed by fitting a least squares function to the data using GraphPrism, and a two-way ANOVA followed by a Sidàk's multiple comparison test was used to determine the significant OR responses. We note that the 4 ORs passed both activation threshold and dose–response threshold.

## Experiment 3

**Participants.** To estimate required sample size, we conducted power analyses on the data obtained in Experiment 1 using G*Power software [61]. At 2 tails, alpha = 0.05 and 80% power, this implied at least 31 participants. Therefore, 33 healthy men (mean age = 27.2 ± 3.3) with no chronic use of medication were recruited for the experiment. All participants provided written informed consent to procedures approved by the Weizmann Institute IRB committee (Protocol # 1514–1) and completed a demographics and AQ questionnaire [59].

**Procedures.** The procedures were identical to Experiment 1, yet in the MRI facility rather than in the behavioral lab, and with the following minor differences: First, we did not conduct 3 test sniffs before the 10 compound sniffs. Second, we deposited 150 μl rather than 100 μl of the stimulus onto the pad pasted under the participant's nostrils. Third, provocation occurred only during the monetary response, as previously done in PSAP-fMRI studies [31,32]. This enables group analysis of brain response to provocation and, critically, assures that the time-window of interest was always when both squeeze balls were pressed, so any reported activations cannot be the result of differences in motor activity. All Experiment 3 procedures lasted around 120 minutes per participant per day, of this around 45 minutes of net scanning time.

**Questionnaires.** As in Experiment 1.

**Statistical analysis—Behavior.** Since there were no blank sniffs, we excluded the first sniff rating of relative descriptors such as familiarity and intensity from each day's analysis. The remainder was done as in Experiment 1.

## MRI data acquisition

MRI scanning was performed on a 3 Tesla Siemens MAGNETOM Prisma scanner, using a 32-channel head coil. Whole-brain functional images were acquired using the T2-weighted Minnesota multiband EPI sequence [62,63] with a multiband acceleration factor of 2, and sequence parameters: 56 slices, TR = 2,000 ms, TE = 30 ms, flip angle = 75°, FOV = 240 × 240 $mm^2$, matrix size = 96 × 96 $mm^2$, voxel size = 2.5 × 2.5 mm, slice thickness = 2.5 mm with no gap. Anatomical images for functional overlay were acquired at 3D T1-weighted magnetization prepared rapid gradient-echo sequence at high resolution: 1 × 1 × 1 $mm^3$ voxel, TR = 2,300 ms, TE = 2.98 ms, inversion time = 900 ms, flip angle = 9°.

## Statistical analysis—fMRI

Functional data were analyzed and processed using FSL 6.0 (FMRIB's Software Library; www.fmrib.ox.ac.uk/fsl), FEAT (FMRI Expert Analysis Tool) v6.00, and MATLAB R2019b (Math-Works). Registration of functional data to high-resolution structural image was carried out using the boundary-based registration (BBR) algorithm [64]. Functional images were spatially normalized to the individual's anatomy and coregistered to the MNI 152 T1 template, using a combination of affine (FLIRT) [65] and nonlinear (FNIRT) [66] registrations. Preprocessing included non-brain removal using BET [67], motion correction using MCFLIRT [68], spatial smoothing using a Gaussian kernel of FWHM 6 mm, grand-mean intensity normalization of the entire 4D dataset by a single multiplicative factor, high pass temporal filtering (Gaussian-weighted least-squares straight line fitting, with sigma = 62.5 seconds). Each participant had 2 runs (one on each day). For each run, a first-level general linear model included the following regressors: revenge (<5 seconds); monetary (<11 seconds); provocation (1 seconds < t < 7 seconds), which was nested within the monetary events. The onset and offset of events were determined as follows: The onset of all events was according to the visual stimulus presented to participants, and so was the offset of the revenge event (Rev.Ev; S1 Fig). The offset of the monetary event was dependent on the participants' behavior. In the case of a nested provocation, the offset of the monetary event was until the onset of provocation (Mon.Ev; S1 Fig), and the offset of provocation (visual stimulus lasted 1 second) was set at the end of the monetary block, dependent on the participant's behavior (Prov.Ev; S1 Fig). The time from provocation onset to the monetary offset following provocation was relatively constant within a participant on both days and for both stimuli (S10 Fig and S5 Table). We regressed out failed events, added temporal derivatives, and regressed out single volumes with excessive motion according to frame-wise displacement > 0.9 mm. The signal was convolved with a double-gamma hemodynamic response function (HRF). The second-level analysis combined both the participant's runs, adding the stimulus for each run accordingly. Analysis was carried out using a fixed effects model by forcing the random effects variance to zero in FLAME (FMRIB's Local Analysis of Mixed Effects) [69,70]. Third-level analysis grouped the data by averaging across all groups. Analysis was carried out using FLAME stage 1 with automatic outlier detection [69,70]. Z (Gaussianised T/F) statistic images were threshold using clusters determined by Z > 2.58 ($p < 0.005$) and a (corrected) cluster significance threshold of $p = 0.05$ [71]. In the analysis, we focused on provocation events, as unlike revenge events, the number of these is relatively consistent across participants [47]. Note that in this design, the contrast of provocation versus inactive time is de facto provocation versus monetary [47].

## Statistical analysis—Psychophysiological interactions analysis (PPI)

We conducted a whole-brain PPI analysis using 2 ROIs that emerged from the group analysis contrast of saline versus tears as seeds to explore functional connectivity with these regions during provocation (left AIC and PFC). Data processing was carried out using FEAT v6.00. The first regressor was the provocation events (psychological regressor), the second was the time-course of the ROI (physiological regressor), and the third was the PPI regressor of the convoluted response (interaction regressor) generated using FSL. Other regressors were the task regressors as in the GLM model (monetary, aggression, none, and motion-outliers according to frame-wise displacement >0.9 mm). The following preprocessing was applied: grand-mean intensity normalization of the entire 4D dataset by a single multiplicative factor. Time-series statistical analysis was conducted using FILM with local autocorrelation correction [72]. Second and group-level analyses were carried out similarly to the GLM model. MNI space gray and white matter was used for pre-threshold masking in the group analysis. Z

(Gaussianised T/F) statistic images were threshold using clusters determined by Z > 2.58 ($p < 0.005$) and a (corrected) cluster significance threshold of $p = 0.05$ [71].

## Exclusions

Behavioral data from 3 participants could not be analyzed due to technical faults of the acquisition system. Additionally, 1 participant reported that he did not believe he played against a real opponent, and another declared he did not have the motivation to engage in the PSAP game on the second day of the experiment. Finally, 2 participants were defined as outliers by the interquartile range (< Q1-1.5 or > Q3+1.5). Thus, 7 participants were not included in the behavioral analysis, 4 of them for exclusions. Finally, 2 participants were excluded from the MRI analyses due to excessive head movements during the scan.

## Supporting information

**S1 Data. S1 Data is an Excel file containing tabs with the data that were used to generate the following figures: Figs 1A–1C, 2A–2C, 2D, 2E, 4B, 4C, 5B–5D, S2D–S2F, S4A–S4C, S6, S7A–S7D, S8A, S8B, S10A and S10B.** The headings on each column allow for orientation. (XLSX)

**S2 Data. S2 Data is an Excel file containing the data from the screening assay for ORs' activation.** The response of 62 ORs to tears is specified in tubs T1-6. The tab named "Tears raw data" summarizes the response of all 62 ORs and the PCI empty vector to tears. The response of 62 ORs to trickled saline is specified in tubs S1-6. The tab named "Saline raw data" summarizes the response of all 62 ORs and the PCI empty vector to tears. In each tab, there is the Luciferase readout (LUC) and the Renilla readout (RL) for tears (in quadruplicate) and for CD293 stimulation medium (in quadruplicate) for each OR. The headings on each column and row allow for orientation. (XLSX)

**S3 Data. S3 Data in a Prism file with the data and statistical analysis of 22 ORs' dose response for tears and saline.** (PZFX)

**S1 Fig. Experimental design.** In a within-participant design, participants were exposed to trickled saline on one day, and tears on the other (counterbalanced for order and double-blind). (**A**) First, participants sniffed the stimulus (tears/saline) from a jar and rated odor perception 10 times. Next, a stimulus-impregnated pad was placed under the participant's nostril for the rest of the experiment. Following that, participants engaged in the (**B**) PSAP game during which they could earn money for themselves (monetary response) or reduce money from their fictitious opponent at no personal gain (revenge response). During the game, they were provoked by money being taken from them by the fictitious opponent (provocation event). (TIF)

**S2 Fig. Tears and trickled saline were not perceptually discriminable from a blank saline solution.** The bar plot depicts the mean ratings of (**A**) pleasantness, (**B**) intensity, and (**C**) familiarity of tears versus trickled saline. Whiskers represent the SE. The scatter plots depict of the normalized VAS ratings of saline solution (blank) and stimulus (tears in blue and trickled saline in red) for (**D**) pleasantness, (**E**) intensity, and (**F**) familiarity. Each dot represents the average of 6 blank sniffs and 10 stimulus sniffs (normalized to min-max values) of each participant. The data in (**D**-**F**) are presented along a unit slope line ($X = Y$), such that if points accumulate above the line, this implies higher values after tears; if points accumulate below the line,

this implies higher values after saline; and if points are distributed around the line, this implies no difference. Data used to generate graphs can be found in S1 Data.
(TIF)

**S3 Fig. Screening for an olfactory receptor response to human tears in vitro.** Screening for olfactory receptor (OR) activation in vitro by 3.16% (v/v) (**A**) tears (blue) and (**B**) trickled saline (red). The response of 62 human ORs was normalized to that of the empty vector (pCI). Bars represent the mean normalized luminescence (Luc/ Rluc), and error bars are standard error (SEM), $n = 4$. For screening only, the luminescence induced by tears/saline was compared to that of the solvent (CD293 medium, in gray) by one-tailed paired $t$ tests. $* = p < 0.05$, $** = p < 0.01$, $*** = p < 0.001$. Data used to generate graphs can be found in S2 Data.
(TIF)

**S4 Fig. The influence of tears on aggression within the MRI scanner.** The behavioral effect of tears in the MRI was subtle, possibly since a day-after-day experiment inside the MR scanner rendered participants significantly more aggressive on the second day, regardless of condition (mean APR difference regardless of condition (day 2—day 1) = 0.511 ±1.2 APR, permutation $p = 0.038$, Mielke and Berry's R = 0.125) as depicted in (**A**). Violin plot of APR score between days, regardless of condition. Thus, participants who sniffed tears on the first day exhibited a remarkable 73.6% lower aggression under tears (mean APR tears day 1 = 0.677 ± 0.942, mean APR saline day 2 = 1.413 ± 1.814, Shapiro–Wilk, W = 0.704, $p < 0.001$, implying a nonnormal distribution dictating a nonparametric test: Wilcoxon signed rank Z = 11, $p = 0.017$, effect size ($r_{rb}$) = 0.758. If we nevertheless use the parametric approach, the effect remains the same: t (14) = 2.34, $p = 0.034$, Cohen's d = 0.605), yet participants who sniffed saline on the first day had no difference in the levels of aggressiveness on the second day under tears (mean APR saline day 1 = 1.159 ± 1.332, mean APR tears day 2 = 1.362 ± 1.751, Wilcoxon signed rank Z = 16, $p = 0.83$, effect size ($r_{rb}$) = 0.111. If we nevertheless use the parametric approach, the effect remains the same: t (10) = 0.579, $p = 0.578$, Cohen's d = 0.175) as depicted in (**B**) and (**C**) respectively. (**B**) Violin plot of APR by day and by stimulus when day 1 was tears. (**C**) Violin plot of APR by day and by stimulus when day 1 was trickled saline. $* = p < 0.05$. Each dot in the violin plots (**B**-**E**) represents a participant. The white dot represents the median, and the gray bar represents the quartiles. Data used to generate graphs can be found in S1 Data.
(TIF)

**S5 Fig. A typical brain response to provocation.** Statistical parametric map of the GLM provocation event, $n = 24$. The color bar represents z-values. $P$ value is depicted. See full list of activated areas in S3 Table. Data are available at https://openneuro.org/datasets/ds004274.
(TIF)

**S6 Fig. Tears' effect remains after counterbalancing for stimuli order.** In the bootstrap analysis, participants were randomly selected 10,000 times to counterbalance for stimuli order. Beta values for (**A**) left AIC and (**B**) PFC were then compared for each stimulus. The histograms represent the distribution of the $t$ statistic of all repetitions for each ROI. The gray horizontal line represents the mean statist. The mean $t$ statists and the $p$-values are depicted. (**C**) Statistical map of the GLM ANOVA Provocation with an added level of saline vs. tears (tears < saline in blue; tears > saline in red), counterbalanced for order, $n = 18$. Z threshold > 2.31, cluster corrected to $p = 0.05$. Color bars represent z-values. Correlation between differences in behavioral APR scores (saline -tears) and differences in beta values (saline- tears) of (**D**) left AIC and (**E**) PFC. Each dot represents a participant. The continuous line represents the fit. The dashed line marks the confidence bounds. Spearman rank

correlation coefficient and *p*-values are depicted. Data used to generate graphs can be found in S1 Data; fMRI data are available at https://openneuro.org/datasets/ds004274. (TIF)

**S7 Fig. Participants' social attitude toward their opponent was not affected by tears.** Participants completed a questionnaire that included social questions regarding their opponent at the end of each experimental day (i.e., once after sniffing tears and once after sniffing saline). Participants answered the following questions using a VAS ranging from "very much" to "not at all": (**A**) Would you like to meet your opponent? (Shapiro–Wilk, W = 0.94, $p < 0.03$, implying a nonnormal distribution dictating a nonparametric test: Wilcoxon signed rank Z = 330, $p = 0.195$, corrected $p = 0.78$). (**B**) Would you like to meet him for beer? (Shapiro–Wilk, W = 0.92, $p = 0.003$, implying a nonnormal distribution dictating a nonparametric test: Wilcoxon signed rank Z = 293, $p = 0.077$, corrected $p = 0.3$). (**C**) Do you think your opponent is a nice person? (t(50) = −0.816, $p = 0.42$, corrected $p > 0.99$). (**D**) How did you feel while playing against your opponent? Participants answered this question using a VAS ranging from "enjoyed it very much" to "Got really angry" (Shapiro–Wilk, W = 0.944, $p < 0.02$, implying a nonnormal distribution dictating a nonparametric test: Wilcoxon signed rank Z = 284, $p = 0.14$, corrected $p = 0.56$). Participants did not show a difference in their social attitude toward their opponent across stimuli. Data used to generate graphs can be found in S1 Data. (TIF)

**S8 Fig. Trait aggression and state anxiety, as measured by AGQ and STAI, respectively, did not correlate with the effect of tears on aggressive behavior.** (**A**) Correlation matrix of AGQ; total score and the different factors with the APR difference between stimuli (Tears–Saline), *n* = 49. The color bar represents the Spearman rank correlation coefficient (r), also depicted. All *p*-values > 0.5 (corrected for multiple comparisons). (**B**) Correlation between STAI score and the APR difference between stimuli (Tears–Saline). Each dot represents a participant, *n* = 47 (2 participants did not completed this questionnaire). The continuous line represents the fit. The dashed line marks the confidence bounds. Spearman rank correlation coefficient and *p*-value are depicted. Data used to generate graphs can be found in S1 Data. (TIF)

**S9 Fig. Calibration of the in vitro assay for OR activation by tears.** We monitored Renilla luciferase luminescence produced by cells (see Methods) transfected with the empty vector negative control (pCI), testing a wide range of stimuli concentrations, tears in blue and trickled saline in red. (**A**) 50%, 25%, and 6.25% (v/v in CD293 stimulation medium). Tears decreased the luminescence, indicating an OR-independent effect on cells at 50% and 25% ($p < 0.001$). At 6.25%, the nonspecific effect was close to significant with $p = 0.0545$. Therefore, we then refined the concentration range to (**B**) 10% to 0.0316% (v/v). The maximum tears concentration without nonspecific OR effects was 3.16%. Statistics are done with a multiple comparison 2-way ANOVA (Dunnett test, $p > 0.05$ = ns; $0.05 < p < 0.01$ = *; $0.01 < p < 0.001$ = **; $p < 0.001$ = ***). Data used to generate graphs can be found in S3 Data. (TIF)

**S10 Fig. Provocation event length was constant for each participant throughout the different days and stimuli.** (**A**) No difference in provocation event length between days (Shapiro–Wilk, W = 0.817, $p < 0.001$, implying a nonnormal distribution dictating a nonparametric test: Wilcoxon signed rank Z = 156, $p = 0.86$, effect size ($r_{rb}$) = 0.16). (**B**) No difference in provocation event length between stimuli, saline in red and tears in blue (Shapiro–Wilk, W = 0.78, $p < 0.001$, implying a nonnormal distribution dictating a nonparametric test: Wilcoxon signed rank Z = 142, $p = 0.95$, effect size ($r_{rb}$) = 0.048). The rectangle reflects the upper and the lower

interquartile (25th to the 75th percentiles), and the whiskers are minimum and maximum non-outlier (no more than 1.5 * IQR of the upper and lower hinges). Outlying points are plotted individually. The line inside the box is the sample median. Each point is a participant, and the line connects the repeated measure. Data used to generate graphs can be found in S1 Data. (TIF)

**S1 Table. Screening for human olfactory receptor activation by tears in vitro.** The table summarizes the screening after OR activation in vitro, of 62 human ORs, induced by tears and trickled saline. The results are depicted in Fig 1. $P$ values of one-tail paired $t$ test comparison between solvent (CD293 medium) to 3.6% (v/v) tears or saline. Activ = considered as activated receptor, Inhib = considered as inhibited receptor. The corresponding $p$-values depicted in the table.
(DOCX)

**S2 Table. OR dose–response statistics.** Repeated measures ANOVA for OR dose–response activation assay. $F$ statistic and $P$ value matched for the compound factor (saline vs. tears).
(DOCX)

**S3 Table. Brain areas activated by provocation events compared to inactive time.** Coordinates and Z-statistics for all significant activation ($P < 0.001$, corrected for multiple comparisons $P < 0.05$) for the contrast Provocation > inactive time.
(DOCX)

**S4 Table. Brain areas activated by provocation events compared to inactive time and saline compared to tears.** Coordinates and max intensity Z-statistics for all significant activation ($P < 0.005$, corrected for multiple comparisons $P < 0.05$) for the contrast Provocation > inactive time and saline > tears.
(DOCX)

**S5 Table. Time from provocation onset to monetary offset.** The mean time from the beginning of provocation event to the end of the monetary response in which it was nested. This time was specified as provocation event in the fMRI GLM analysis for each subject. Time in seconds of both sessions (day 1 and day 2) are depicted.
(DOCX)

## Acknowledgments

We thank Dr. Aharon Ravia and Dr. Ofer Perl for their advice.

## Author Contributions

**Conceptualization:** Shani Agron, Hiroaki Matsunami, Noam Sobel.

**Data curation:** Shani Agron, Claire A. de March, Reut Weissgross, Eva Mishor, Tali Weiss, Edna Furman-Haran, Hiroaki Matsunami.

**Formal analysis:** Shani Agron, Claire A. de March, Lior Gorodisky, Tali Weiss, Hiroaki Matsunami, Noam Sobel.

**Funding acquisition:** Hiroaki Matsunami, Noam Sobel.

**Investigation:** Shani Agron, Claire A. de March, Reut Weissgross, Eva Mishor, Lior Gorodisky, Tali Weiss, Hiroaki Matsunami.

**Methodology:** Shani Agron, Claire A. de March, Reut Weissgross, Eva Mishor, Lior Gorodisky, Tali Weiss, Edna Furman-Haran, Hiroaki Matsunami, Noam Sobel.

**Project administration:** Shani Agron, Hiroaki Matsunami, Noam Sobel.

**Resources:** Hiroaki Matsunami, Noam Sobel.

**Software:** Shani Agron.

**Supervision:** Hiroaki Matsunami, Noam Sobel.

**Validation:** Shani Agron, Noam Sobel.

**Visualization:** Shani Agron.

**Writing – original draft:** Shani Agron, Noam Sobel.

**Writing – review & editing:** Shani Agron, Claire A. de March, Reut Weissgross, Eva Mishor, Lior Gorodisky, Tali Weiss, Hiroaki Matsunami, Noam Sobel.

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
