## [Editor Report · Decision Letter 0]

15 May 2023

Dear Dr Sobel, 

Thank you for submitting your manuscript entitled "Tears Provide a Chemical Blanket Protecting from Aggression" for consideration as a Research Article by PLOS Biology.

Your manuscript has now been evaluated by the PLOS Biology editorial staff as well as by an academic editor with relevant expertise and I am writing to let you know that we would like to send your submission out for external peer review.

Once your full submission is complete, your paper will undergo a series of checks in preparation for peer review. After your manuscript has passed the checks it will be sent out for review. To provide the metadata for your submission, please Login to Editorial Manager (https://www.editorialmanager.com/pbiology) within two working days, i.e. by May 17 2023 11:59PM.

As a note - after some discussion in the team, we think your manuscript would be best suited for our Short Report format. We therefore request you change the article type to Short Report, as you complete your submission. For more information on Short Reports, see here: https://journals.plos.org/plosbiology/s/what-we-publish#loc-short-reports

Kind regards,

Lucas

Lucas Smith, Ph.D.

Associate Editor

PLOS Biology

lsmith@plos.org

---

## [Decision Letter · Decision Letter 1]

30 Jun 2023

Dear Dr Sobel,

Thank you for your patience while your manuscript "Tears Provide a Chemical Blanket Protecting from Aggression" was peer-reviewed at PLOS Biology. It has now been evaluated by the PLOS Biology editors, an Academic Editor with relevant expertise, and by several independent reviewers who are generally positive about the study, but who have a number of comments and concerns that will need to be addressed. In light of the reviews, which you will find at the end of this email, we would like to invite you to revise the work to thoroughly address the reviewers' reports.

Given the extent of revision needed, we cannot make a decision about publication until we have seen the revised manuscript and your response to the reviewers' comments. Your revised manuscript is likely to be sent for further evaluation by all or a subset of the reviewers.

**IMPORTANT - SUBMITTING YOUR REVISION**

*Re-submission Checklist*

*Published Peer Review*

*PLOS Data Policy*

*Blot and Gel Data Policy*

Sincerely,

Luke

Lucas Smith, Ph.D.

Senior Editor

PLOS Biology

lsmith@plos.org

REVIEWS:

Reviewer #1, Keith M Kendrick (note, Reviewer 1 has signed this review): 

This is an intriguing follow up study by this group reporting that exposure to the emotional tears of women reduces aggressive behavior in men during performance of an economic game. Their behavioral findings are complimented by additional ones showing that some olfactory receptors can respond dose dependently to the womens' tears in vitro and that reduced activity occurred in some olfactory/aggression processing brain regions when men were exposed to the tears in an MRI scanner. Overall, the study is interesting and thorough as well as technically challenging and the paper is well written. I do have some comments however which I think the authors should address:

I think that to state the tears are odorless is technically incorrect since they appear to be able to influence the olfactory system and the whole claim is that the reduction in aggression is due to the olfactory cues in the tears. What does appear to be true is that the odors from the tears themselves cannot be consciously discriminated by human males. This is perhaps somewhat akin to subliminal processing of visual cues which can influence behavior but without being consciously perceived.

I find myself struggling a little to find a context in which the odor of tears might influence a man's behavior given that in humans both visual and auditory cues for crying are far easier for us to perceive. It is much simpler to conceptualize this type of signaling in nocturnal animal species where individuals live in close proximity to one another, and of course also have a functional vomeronasal system. I think it would be useful if the authors at least speculate in what contexts the odor of women's tears might be relevant. At this point we also don't know if the sight of tears or the sounds of crying might produce similar aggression reducing effects.

The authors measured state anxiety and trait aggression using STAI and Buss-Perry questionnaires after the experiment. It would be useful to know to what extent these measures correlate with aggression scores on the PSAP. Stimuli which evoke aggression often do so mainly in individuals who have high trait aggression so it would be interesting to know if the effects of tears in reducing aggression were greatest in individuals with higher trait aggression.

The authors excluded one subject as a statistical outlier based on the standard definition of an outlier using IQ ranges, however they also used non-parametric statistics due to non-normal distribution which generally makes data analysis insensitive to outliers, at least in terms of type 1 errors. Does inclusion of this subject prevent achievement of a significant p value?

Minor point: On page 15, line 9 "immerged" should be "emerged"

Reviewer #2: 

General comments

The authors tested the influence of human tears on behavioral aggression, olfactory receptors, and fMRI activity in aggression networks. The approach is innovative and yields intriguing results in an area that is so far understudied in humans. However, there are some significant issues that the authors should address to increase confidence in the results. The biggest sticking point is related to experiment 3, where there are concerns about the way the behavioral results are reported, and a significant imbalance across conditions following exclusions. Other concerns are mainly related to the narrative structure of the article, and errors/inconsistencies/gaps in reporting, which are more easily addressed. 

Specific comments

Major comments

* The introduction focuses on biological mechanisms of chemical signaling (e.g., the role of ESP1), which is interesting and somewhat relevant. However, given the nature of the study, shifting the introduction's focus toward behavioral outcomes, especially in humans, would provide helpful context. For instance, the authors could briefly report behavioral and fMRI findings from Gelstein et al., 2011 (from the same group), Mishor et al., 2021 (also from the same group), and Oh et al., 2012, and discuss the known fMRI correlates of aggression. 

* Behavioral findings from experiment 3 should be reported in a similar manner to those from experiment 1 (i.e., by comparing APR values across conditions for the entire group, and plotting the data in figures similar to those in Figure 1 [D-F]). Although it's plausible that order effects could be accentuated in the MRI scanning context, properly counterbalancing the conditions should account for this issue. However, based on data file 4, it seems that the groups were no longer well-balanced following exclusions, with 11 participants in the saline-first group, and 15 participants in the tears-first group. If order plays a significant role in modulating aggression as the authors suggest, this is a significant problem, since differences in fMRI correlates of aggression could be driven by the order of conditions, rather than the chemical stimulus applied. Ideally, the authors would recruit more participants to balance the groups, or at least account for the imbalance somehow in their models. 

* Although results from this study could be related to COVID-19, this is very speculative. It's not clear that anosmia in COVID patients results in a parallel deficit in "odorless" chemical signaling, and there are a host of other (probably more salient) reasons why aggression might manifest in individuals with a complex illness. Drawing conclusions about men in particular seems inappropriate, given that there's no gender comparison in this study. Perhaps the authors could bring up the COVID-19 link, but in a more measured way, and without emphasizing it. 

Minor comments

* Were participants asked whether they believed their PSAP opponent was human? If not, this should not be assumed in the results section.

* It's stated that participants did not know they were sniffing tears during the sniff-jar procedure. What were the participants told, if anything, about the stimulus?

* Could the authors include bar plots to illustrate the mean pleasantness, intensity, and familiarity for tears vs. saline in Figure 1? There's also an inconsistency on the plot (which references experiments 1 and 2) and the legend and supplementary data file (which reference experiments 2 and 3) - from reading the manuscript, I believe that all of these should reference experiments 1 and 3. Most importantly though, the idea that the tears are "odorless" is not supported by the data, since participants rate the tear stimuli as moderately intense. It would be more accurate to say that previous studies (e.g., Gelstein et al., 2011) have demonstrated that tear and saline stimuli cannot be discriminated based on odor, which implies that the chemical signal contained in tears is odorless. The data collected here support this idea (since there aren't meaningful differences in ratings across stimuli), but it is not tested directly. 

* The APR is a ratio, but several participants have a score of 0 for one or both conditions. What does a 0 value mean in this case? And do the numbers following the mean APR (e.g., 1.67+/-1.7 APR) represent the standard error?

* It does not seem appropriate to include OR1A1 in the olfactory receptor analysis, given that it does not meet the established criteria, and there are other ORs that are similarly close to the cutoff, but are not included (e.g., OR5K1). If the authors feel that it's important to include this receptor, then they should explain why in the main text (beyond stating general interest), and this should be framed as a separate exploratory analysis (after the main analysis). Could the authors also provide justification for moving forward with receptors that are activated, but not inhibited, by the stimuli (given that inhibition could also occur in a dose-dependent manner)?

* The authors state that the order of conditions was counterbalanced for experiment 1, and time of day was held constant. Can the authors clarify (in the main text) whether this was the case for experiment 3 as well?

* The authors state: "The contrast of provocation vs. baseline regardless of condition revealed a typical salience network activation, which included typical provocation-associated regions." Framing the control condition as a "baseline" seems counterintuitive, since it implies that there's no contrast. Can the authors use another label for the control condition in the main text and associated figures to avoid confusion? 

* The authors should add p-value thresholds to MRI figure legends and cluster sizes to supplementary MRI tables for more complete information. 

* The authors should make it clear whether the difference metrics represented in Figure 3 B-C are subtracting the saline values from tears values, or vice versa (so the direction of the correlation is well-defined). 

* Was the olfactory cluster the only cluster to emerge in the functional connectivity analysis? If so, this should be clarified. If not, the authors should provide a supplementary table that lists the brain regions observed (as they did for prior analyses). 

* Although results suggest that neural responses are decreased in the tear condition, and functional connectivity with an olfactory-adjacent region is increased, the authors should refrain from implying that one result causes the other (e.g., "We also find that this signal increases functional connectivity between the brain substrates of olfaction and aggression, reducing overall levels of activity in the aggression brain network.")

* Why were participants exposed to "blanks" (saline?) for 3 sniffs prior to the 10 main sniffs in experiment 1, and then not in experiment 2? 

* For the various questionnaires the participants completed, could the authors report the rationale for administering these and/or the main results? For instance, participants completed an autism quotient questionnaire - why? Participants also completed anxiety and aggression questionnaires, and rated how much they would want to meet their opponent. Reporting the difference (or lack of) across conditions could be informative. 

* Participants also donated saliva. Why did they do this? Was the testosterone content analyzed? 

* Given that participants sniffed "blanks" prior to the 10 main sniffs in experiment 1, it does not seem necessary to remove the first sniff rating from the analysis. Could the authors report whether the results change if the first rating is included? There are also contradictions here, since the ratings were normalized to min-max values within all twenty odor exposures, and Figure 1 references the average of ten sniffs by a given participant. 

* For participants who were excluded for being outliers, was this based on their APR scores? Could the authors report these scores (at least in the methods and data files, and ideally in the main figure legends) for completeness? It's also unclear why a participant would be excluded for reporting that he did not believe he played against a real opponent. Again, was this probed specifically? Or based on an offhand comment? If it's the latter, it's possible that other participants felt similarly, even though they did not express it. 

* Could the authors clarify the timeline for the PASP in experiments 1 and 3? In the main text it's implied that this took 2 hours, but the authors mention a single MRI run (per day) in the methods. 

* Could the authors report whether there are statistically significant differences in mean time from provocation onset to monetary offset for day 1 vs. day 2, and for tears vs. saline? 

* The methods reference regressing out volumes according to frame-wise displacement of greater than 0.9. Is this 0.9 mm?

* The authors state that they conducted whole-brain PPI analyses using three ROIs as seeds in the methods, but then only name two of these ROIs (left AIC and PFC). 

Reviewer #3: 

In this manuscript, the authors set out to test the effect of the smell of tears on aggression. The authors collect tears from females watching sad film, and assess the effect of smelling sad tears on the level of aggression male participants demonstrate in a monetary computer game. The authors observe a decrease in aggressive behavior when participants smell tears, relative to 'trickled saline' controls. In vitro reporter assays show that tears can activate human odorant receptors expressed in cultures cells. Finally, the authors use fMRI to identify brain areas that are active during aggressive behavior in the computer game, and whose activity is modulated by tears. The analyses of imaging data suggests that tears increase functional connectivity between the left anterior insular cortex, previously implicated in aggression, and the right amygdala and piriform cortex, involved in olfactory processing. 

This manuscript extends earlier findings, including by the same lab, that sad female tears contain chemosignals that can alter male testosterone levels and emotional behavior. The use of a monetary computer game to measure aggression, and the finding that tears can activate human odorant receptors in vitro provide important new information towards understanding the relevance of human tearing and its underlying brain mechanisms. With minor revisions (see below), the manuscript is a good fit for publication in PLoS. 

1) The authors highlight the role of emotional tears, but they do not provide any evidence that the effects they observed are selectively caused by emotional tears. Can the authors use e.g. reflex tears as controls for aggressive behavior and odorant receptor signaling? If so, such experiments could provide strong additional support for their conclusions. 

2) Figure 1A-C are difficult to read and interpret. The clouds of dark and light gray dots are difficult to evaluate - can the authors provide a visualization and quantification that allows to connect experiments 1 and 2? Also, if tears are odorless, how is the reader expected to interpret variability in pleasantness, intensity, and familiarity? 

3) The in vitro odorant receptor signaling assays the authors use tend to rely on high odorant concentrations, often substantially higher than used in behavioral tests. Here, tears are used undiluted in behavioral tests, but diluted ~1:30 for odorant receptor signaling assays. The authors should comment on what they estimate are physiologically relevant concentrations of odorant receptor ligands in tears. 

4) For aggression behaviors during fMRI, the authors suggest that overall aggression is increased on day 2, that upon exposure to tears on day 1, this increase is amplified, and that upon exposure to tears on day 2, this increase is attenuated or eliminated. The authors should perform their statistical tests to specifically address the significance of these comparisons. 

5) The manuscript, especially the introduction, appears to heavily focus on earlier work in rodents as a motivation for experiments in humans. The manuscript may benefit from a more explicit discussion of the role of human olfaction including for emotional behaviors, and its link to hormonal regulation. 

6) What do the authors mean by 'capture a genuine link between brain and behavior'?

---

## [Decision Letter · Decision Letter 2]

3 Nov 2023

Dear Dr Sobel,

Thank you for your patience while we considered your revised manuscript "Tears Provide a Chemical Blanket Protecting from Aggression" for consideration as a Short Report at PLOS Biology. Your revised study has now been evaluated by the PLOS Biology editors, the Academic Editor and the original reviewers. The reviewer comments are either appended below, or attached to this email.

As you will see in their comments, both reviewers 1 and 3 are fully satisfied by the revision. However Reviewer 2 has a number of lingering concerns which we think should be addressed. Additionally, we have a number of editorial requests, which I have appended those below my signature. 

In light of the reviews we are pleased to offer you the opportunity to address the remaining reviewer and editorial requests in a revision that we anticipate should not take you very long. However, if you need extra time, please do let us know - we would be happy to provide you with an extension, as needed. 

We will then assess your revised manuscript and your response to the reviewers' comments with our Academic Editor aiming to avoid further rounds of peer-review, although might need to consult with the reviewers, depending on the nature of the revisions.

**IMPORTANT - SUBMITTING YOUR REVISION**

*Resubmission Checklist*

*Published Peer Review*

Sincerely,

Luke

Lucas Smith, Ph.D.

Senior Editor

PLOS Biology

lsmith@plos.org

REVIEWS:

Reviewer #1, Keith M Kendrick (note, Reviewer 1 has signed this review): The authors have carefully revised and improved their manuscript in response to comments made

Reviewer # 2 (see attached): 

Reviewer #3: This is a carefully revised and improved manuscript. The authors addressed all my concerns. I recommend publication in PLoS Biol.

EDITORIAL REQUESTS: 

1) TITLE: After some discussion within the team, we think 'chemical blanket' may be a bit too metaphorical for the title, and so we would like to suggest that the title be edited for clarity. We suggest that you change it to something like: 

"A chemical signal in female tears lowers aggression in men"

We are happy for you to optimize this further.

2) DATA: I see in your data availability statement that you plan on making your MRI data available on OpenNeuro. Can you please provide me a reviewer link so that I can verify that it meets our requirements for data sharing?

3) DATA: In addition to providing the raw imaging data related to your study, you may be aware of the PLOS Data Policy, which requires that all data be made available without restriction: http://journals.plos.org/plosbiology/s/data-availability. For more information, please also see this editorial: http://dx.doi.org/10.1371/journal.pbio.1001797

a. Supplementary files (e.g., excel). Please ensure that all data files are uploaded as 'Supporting Information' and are invariably referred to (in the manuscript, figure legends, and the Description field when uploading your files) using the following format verbatim: S1 Data, S2 Data, etc. Multiple panels of a single or even several figures can be included as multiple sheets in one excel file that is saved using exactly the following convention: S1_Data.xlsx (using an underscore).

b. Deposition in a publicly available repository. Please also provide the accession code or a reviewer link so that we may view your data before publication. 

>>Regardless of the method selected, please ensure that you provide the individual numerical values that underlie the summary data displayed in the following figure panels as they are essential for readers to assess your analysis and to reproduce it:

Fig 1A-F; Fig 2A-E; Fig 3B-C; Fig 4B-D; FIg S2A-F; FIg S3A-B; Fig S4A-E; FIg S5A-B,D-E; Fig S7A-D; Fig S8B Fig S9A-B; FIg S10A-B

>>Please also ensure that figure legends in your manuscript include information on where the underlying data can be found, and ensure your supplemental data file/s has a legend.

---

## [Editor Report · Decision Letter 3]

21 Nov 2023

Dear Dr Sobel,

Thank you for the submission of your revised Short Report, "A chemical signal in human female tears lowers aggression in males" for publication in PLOS Biology, and thank you for addressing the last reviewer and editorial requests in this revision. On behalf of my colleagues and the Academic Editor, Thorsten Kahnt, I am pleased to say that we can in principle accept your manuscript for publication, provided you address any remaining formatting and reporting issues. These will be detailed in an email you should receive within 2-3 business days from our colleagues in the journal operations team; no action is required from you until then. Please note that we will not be able to formally accept your manuscript and schedule it for publication until you have completed any requested changes.

PRESS

Sincerely, 

Lucas Smith, Ph.D.

Senior Editor

PLOS Biology

lsmith@plos.org